# Plasticity in the Morphology of Growing Bamboo: A Bayesian Analysis of Exogenous Treatment Effects on Plant Height, Internode Length, and Internode Numbers

**DOI:** 10.3390/plants12081713

**Published:** 2023-04-20

**Authors:** Chongyang Wu, Yucong Bai, Zhihua Cao, Junlei Xu, Yali Xie, Huifang Zheng, Jutang Jiang, Changhong Mu, Wenlong Cheng, Hui Fang, Jian Gao

**Affiliations:** 1Key Laboratory of National Forestry and Grassland Administration, Beijing for Bamboo & Rattan Science and Technology/International Center for Bamboo and Rattan, Beijing 100102, China; wcy@icbr.ac.cn (C.W.); bai.yucong@icbr.ac.cn (Y.B.); xjl@icbr.ac.cn (J.X.); xieyali@icbr.ac.cn (Y.X.); jiangjutang@icbr.ac.cn (J.J.); muchanghong@icbr.ac.cn (C.M.); chengwenlong@icbr.ac.cn (W.C.); fanghui@icbr.ac.cn (H.F.); 2Anhui Academy of Forestry, Hefei 230036, China

**Keywords:** bamboo, plant morphology, sucrose, gibberellin, Bayesian analysis, plasticity

## Abstract

Sucrose (Suc) and gibberellin (GA) can promote the elongation of certain internodes in bamboo. However, there is a lack of field studies to support these findings and no evidence concerning how Suc and GA promote the plant height of bamboo by regulating the internode elongation and number. We investigated the plant height, the length of each internode, and the total number of internodes of Moso bamboo (*Phyllostachys edulis*) under exogenous Suc, GA, and control group (CTRL) treatments in the field and analyzed how Suc and GA affected the height of Moso bamboo by promoting the internode length and number. The lengths of the 10th–50th internodes were significantly increased under the exogenous Suc and GA treatments, and the number of internodes was significantly increased by the exogenous Suc treatment. The increased effect of Suc and GA exogenous treatment on the proportion of longer internodes showed a weakening trend near the plant height of 15–16 m compared with the CTRL, suggesting that these exogenous treatments may be more effective in regions where bamboo growth is suboptimal. This study demonstrated that both the exogenous Suc and GA treatments could promote internode elongation of Moso bamboo in the field. The exogenous GA treatment had a stronger effect on internode elongation, and the exogenous Suc treatment had a stronger effect on increasing the internode numbers. The increase in plant height by the exogenous Suc and GA treatments was promoted by the co-elongation of most internodes or the increase in the proportion of longer internodes.

## 1. Introduction

Bamboos (Poaceae, Bambusoideae) can be divided into woody and herbaceous species based on their culm and flowering characteristics and are one of the most important forest resources in the world [1,2]. Woody bamboo has a higher economic value than herbaceous bamboo since bamboo materials and shoots are more valuable. The economic value of bamboo is directly related to its culm height, and thus, culm height growth is an aspect of bamboo research. The woody bamboos are distributed in tropical and subtropical regions. Owing to their high ecological and economic value [3,4,5,6,7], woody bamboos are widely cultivated in many countries outside their natural range [8,9].

China may be the origin region of bamboo. At present, China has the world’s most diverse bamboo resources [10]; there are 534 bamboo species in 34 genera [11]. The bamboo forest area covers 64,116 km^2^ [12]. China is the world’s largest producer, consumer, and exporter of bamboo products. In 2020, the total import and export trade of bamboo products in China was USD 2.21 billion [13]. Moso bamboo (*Phyllostachys edulis*) is a perennial woody plant that covers an area of 46,778 km^2^, accounting for 72.96% of the total bamboo forest area [12]. This species is the most crucial dual-purpose plant, producing bamboo shoots and material.

Bamboo material has excellent rigidity, hardness, flexural strength, and ductility, and it is widely used as a material for engineering and in daily applications [7,14,15]. Bamboo-flattening technology is considered one of the most advanced and mainstream technologies. This technique uses intact internodes for pressure flattening or planing flattening; as a result, longer internodes can add higher value to bamboo products, such as flooring, cutting boards, and furniture [16,17,18,19]. Meanwhile, from an ecological perspective, longer internodes mean increased biomass, leading to higher bamboo height, thicker culms, and more branches and leaves, which are features that are directly related to the carbon sinks of bamboo forests [20,21]. Although longer internodes are important to bamboo processing and ecology, how to increase the internodes at a low cost is an unsolved problem.

Sugar (e.g., sucrose, glucose, fructose) is the energy source within plants; sucrose (Suc) is not only the main photosynthetic product but is also the main form of carbohydrate storage in many higher plants [22,23,24]. The Suc is transported through the phloem to tissues and organs that do not perform photosynthesis, and it plays a vital role in various stages of plant growth [24,25,26]. In a study of tomato and sorghum, it was found that Suc was closely related to the growth of axillary buds, and Suc affects the length of lateral branches by regulating some development-related genes [27,28]. In other studies, exogenous Suc treatment was found to promote the growth of cucumber seedlings [29] and the root growth of rice seedlings [30]. Different concentrations of Suc can induce stomatal opening or closing in *Arabidopsis thaliana* [31]. Although the external application of Suc can promote plant growth, excessive Suc concentration can cause changes in osmotic pressure during growth, resulting in plant death [32,33]. Sugar also plays a vital role in the rapid growth of bamboo. The catabolism of starch and soluble sugars was significantly increased in the spring shoots of *P. edulis* [34]. The culm sheath uses water as a medium to transport organic acids, sugars, and other compounds to the internodes for the growth of *P. edulis* shoots [35]. Wang et al. [36] found that sugar content was high in the fast elongation and mature internodes of *Fargesia yunnanensis*, implying that the elongation of bamboo internodes requires a large amount of sugar. Several other studies further supported this conjecture. Wei et al. [37] applied 5 g L^−1^ Suc to *Bambusa multiplex* and found positive correlations between Suc content and the elongation rate of internodes. A study of *F. yunnanensis* showed that the starch content in shoot buds was inadequate for the consumption of bamboo shoots during the rapid growth period, and more carbohydrates needed to be obtained from the parent plant [36]. Suc could not be detected until the end of the rapid growth period of *Fargesia yunnanensis*, and the Suc transported to the internodes was entirely hydrolyzed, suggesting that the energy in the bamboo shoots may not be sufficient to fully support the rapid elongation of the internodes [36]. All of the above results show that sugar is highly correlated with the rapid growth of bamboo. Is it possible to increase the internode length by supplementing plants with sugar during the rapid growth period?

Gibberellin (GA) was first discovered from the pathogenic fungus *Gibberella fujikuroi* of *Oryza sativa*, and this discovery triggered the first Green Revolution [38]. GA has many roles in the developmental processes of plants, including seed germination [39] and flower and fruit formation and development [23,39,40,41]. It is noteworthy that the most apparent effect of GA is to regulate stem length and internode elongation during the growth period by inducing cell division and elongation [42,43]. This was demonstrated in model plants and crops, such as *Arabidopsis* [44,45], rice [46], maize [23,47], Napier grass [48], and *Medicago truncatula* [49]. GA is primarily used for dwarfing in agriculture, and there have been many reports on this in crops, such as wheat [50,51,52,53], rice [54,55], maize [56,57], and watermelon [58,59]. Crops need to be dwarfed by GA to increase yield. In contrast, bamboo requires greater plant height to increase economic and ecological value. When the shoot height of *P. edulis* exceeds 1.5 m, the GA content in the middle of the shoot is higher than in the tip and bottom [60]. GA signaling may affect the growth of *P. edulis* by regulating gibberellic-acid-stimulated transcript genes or general regulatory factor proteins [61,62]. However, the effects of GA on bamboo internodes have been less well studied. Wang et al. [63] reported that *P. edulis* f. *tubaeformis* (a dwarf variant of Moso bamboo) has shorter internodes and plant heights than wild Moso bamboo, and the GA content in the internodes of the variant is also lower than that of wild Moso bamboo. Similar results were found in *P. edulis* seedlings; the seedlings under GA treatment had longer internodes and higher plant height [64], faster growth rates, and longer internode cells than the controls [65].

To date, studies suggest that Suc and GA may promote plant height and internode elongation in bamboos [37,63,64,65]. However, most studies focused on specific internodes or seedlings of bamboo, and there is no field investigation evidence concerning the changes in plant height or internode length under exogenous Suc and GA treatments. Because plant height and internode length are very important to the economic and ecological value of bamboo, focusing on the effects of Suc and GA on plant height and internode length of *P. edulis*, we posed the following questions:(1)Does exogenous Suc or GA treatment promote the plant height of *P. edulis* in the wild?(2)Is it a large elongation of a few internodes or a small elongation of all the internodes that promotes an increase in the plant height of *P. edulis*?(3)Does having more internodes promote the increase in plant height of *P. edulis*?(4)Does the promotion of internode length under exogenous Suc and GA treatments diminish with distance from the injection point?

In this study, we subjected *P. edulis* in a wild bamboo forest to exogenous Suc and GA treatments and measured the plant height, internode length, and internode number under different treatments. Using Bayesian modeling to analyze the effects of exogenous Suc and GA on plant height, internode length, and internode number, we discuss how Suc and GA promote the plant growth of *P. edulis* in view of the above four questions.

## 2. Results

### 2.1. Effects of Exogenous Treatments on the Plant Height, Internode Numbers, and Internode Lengths

Exogenous Suc and GA treatments resulted in significantly higher plant heights than in the CTRL (Figure 1a). The observed values of plant height under exogenous Suc treatments were more concentrated than others, and no *P. edulis* was smaller than 10 m in all 79 samples. There were more internodes under exogenous the Suc treatments than in the CTRL, and there were no significant differences between exogenous GA treatments and the CTRL (Figure 1b). The internode length was significantly higher under exogenous Suc and GA treatments than in the CTRL (Figure 1c–f). Details of the statistical results are shown in Appendix A.

### 2.2. Data Characteristics and Correlations between Variables

The multi-density plots in Figure 2 show the distribution of each variable under the CTRL, exogenous Suc, and GA treatments. The distributions of plant heights in different treatments were similar, but the data in the exogenous Suc treatment were concentrated around 16 m. There were more plants higher than 18 m in the exogenous GA treatment. The distribution of internode numbers coincided in the CTRL and exogenous GA treatments, and the distribution was shifted to the right in the exogenous Suc treatment. For the lengths of the 10th–50th internodes in the exogenous Suc and GA treatments, the distribution was shifted to the right and was more concentrated around 0.30–0.35 m compared with the CTRL. The numbers in Figure 2 show the Pearson correlation analysis. There were strong correlations between the plant height and the lengths of the 30th–40th internodes, plant height and the lengths of the 40th–50th internodes, the lengths of the 20th–30th internodes and the lengths of the 10th–20th internodes, and the lengths of the 20th–30th internodes and the lengths of the 30th–40th internodes (r > 0.8, *p* < 0.001). However, in comparing all variables, the correlations in the CTRL treatment were higher than those of the exogenous Suc and GA treatments in almost all cases (Figure 2).

### 2.3. Effects of Exogenous Treatments on the Internode Lengths

Figure 3 shows the differences in the internode length under the CTRL, exogenous Suc, and GA treatments. The internode length increased from the base, reached a peak near the 25th–30th internode, and then gradually decreased until the tip of the tree. Exogenous Suc and GA treatments resulted in significantly longer total lengths of the 10th–20th, 20th–30th, 30th–40th, and 40th–50th internodes than the CTRL. The length increase in the useful internodes (>20 cm) was approximately 1.2 m in the exogenous Suc treatment and approximately 1.4 m in the exogenous GA treatment. The details of these statistical results are shown in Appendix A.

### 2.4. Variation in Promotion of Internode Elongation under Exogenous Treatments with Distance

To know whether the promoting effect of internode length under different exogenous treatments varied with distance, we examined the interaction effect between exogenous treatments and distance from the injection point. The interaction effect between different exogenous (CTRL–Suc) treatments and the internode number was not significant (Figure 4, Appendix A). This result means that the effect of exogenous Suc treatment on internode length did not enhance or attenuate with increasing distance from the injection point. However, the interaction effect between different exogenous (CTRL–GA) treatments and internode number was significantly positive in the 10th–20th and significantly negative in the 20th–30th internodes and was not significant in the 30th–40th or 40th–50th internodes (Figure 4, Appendix A).

### 2.5. Effects of Exogenous Treatments on the Percentage of Internodes with Different Lengths

The proportion of 30–40 cm internodes was significantly higher under exogenous Suc treatment compared with the CTRL, but there was no significant difference in other lengths of internodes (Figure 5, Appendix A). The results indicate that the increasing internode numbers under exogenous Suc treatment were largely from the 30–40 cm internodes. The proportions of 10–20 cm and 20–30 cm internodes were significantly lower, and those of the 30–40 cm and 40–50 cm internodes were significantly higher in the exogenous GA treatment compared with the CTRL (Figure 5, Appendix A). The decreasing numbers with short internodes counterbalanced the increasing numbers with long internodes. Under exogenous Suc and GA treatments, increases were seen in the long internodes; the longer the internodes, the greater their utility for applications.

### 2.6. Effects of Exogenous Treatments on the Percentages of Internodes at Different Plant Heights

Figure 6 and Appendix A show the interaction effects between different exogenous treatments and plant height to further confirm whether plant height affected the exogenous hormonal effects on the proportions of internode lengths. We found similar trends between the exogenous Suc and GA treatments. The proportion of 0–10 cm internodes was significantly higher, and the proportions of 20–30 cm, 30–40 cm, and 40–50 cm internodes were significantly lower than in the CTRL with increasing plant height. Judging from the intersection of the 95% Bayesian credible intervals, the increased effect of the exogenous treatments on the proportion of longer internodes (>20 cm) showed a weakening trend near the plant height of 15–16 m compared with the CTRL.

### 2.7. Effects of Exogenous Treatments on Ecological Benefits

To evaluate the effect of exogenous treatments on the ecological benefits of *P. edulis*, we estimated biomass and carbon storage per plant for different exogenous treatments. The biomass and carbon storage were significantly higher under the exogenous Suc and GA treatments than in the CTRL, and under the exogenous Suc treatment, the increases were approximately 1.2-fold; under the exogenous GA treatment, the increases were approximately 1.4-fold higher than under the CTRL (Figure 7, Appendix A).

## 3. Discussion

In the present study, we measured the plant height, internode length, and internode numbers of *P. edulis* under exogenous Suc and GA treatments. Owing to the field experiment of large bamboos requiring significant resources and manpower, this is the first published report demonstrating the effects of exogenous Suc and GA treatments on the plant height, internode length, and internode number of *P. edulis*. Our results show that both Suc and GA had a significant positive effect on the plant height; Suc had a stronger effect on internode number compared with GA, while GA had a stronger effect on internode length than Suc. In addition, we did not find the greater number of internodes (75) or higher plant height (21.0 m) of *P. edulis* that were published previously [63,65,66,67,68,69,70,71,72,73,74]. Here, the characteristics of the culm structure, the promoting effects of the exogenous Suc and GA on the plant height, internode length, and internode number, the variation in the effect of exogenous treatment with distance, and the ecological benefits brought by increased plant height are discussed. These also cover the four questions raised when presenting the objectives of this study.

### 3.1. Exogenous Treatments Did Not Affect the Characteristics of the Culm Morphological and Internode Length Distributions

Our results showed that the distribution of internode length in *P. edulis* had distinct features: gradually increasing from the bottom, peaking near the 25th–30th internode, then gradually getting shorter again, showing a shape similar to a Gaussian distribution (Figure 3). These results are in accord with those of recent studies [65], which indicated that the diameter of the internodes gradually decreases from the bottom to the top, becoming extremely thin at the tip; the culm shape is similar to a cone [68,74,75]. *P. edulis* is generally taller than 10 m, and some individuals reach about 20 m [70,71,72,73,76]. This height is rapidly achieved within a few months during the rapid growth period, and there is no doubt that the internodes at the bottom are subjected to greater mechanical stress compared with the tip. This mechanical stress may be one of the reasons for the shortening of the internode length at the bottom, where lignification has not yet been completed [37]. The mechanical stress effect in plant development is a general phenomenon [37,77,78,79,80].

Additionally, we considered that the short internodes at the tip may be related to the lack of water and nutrients. The height of *P. edulis* increases rapidly, but the emergence of branches and leaves does not appear during the rapid growth period [81,82]; without leaf transpiration, it may be difficult to transport water to internodes at the tip simply using root pressure [83,84,85]. It is well known that water can be used as a vehicle to transfer carbohydrates and nutrients [81,86,87]. Accordingly, we can infer that the internodes closer to the tip may lack water and nutrients, causing the internodes to be thin and short during the growing stage. Therefore, the mechanical stress on the internodes of the bottom and the nutritional deficiency of internodes at the tip may be why the internode length presented a Gaussian distribution and the culm was in the shape of a cone. However, this hypothesis lacks empirical support. For example, we can control the pressure by truncating the upper internodes and then measuring the internode length under different pressures during the rapid growth period. Alternatively, water transport and nutrient content of different internodes (bottom, middle, and tip) can be measured. Our results indicated that the exogenous Suc and GA treatments did not significantly alter the characteristics of culm morphology or node length distribution.

### 3.2. Effects of the Exogenous Treatments on the Internode Lengths

Our analysis showed that the GA treatment significantly promoted internode elongation (Figure 1c–f and Figure 3b). These results were in accord with recent studies indicating that GA affects internode length by stimulating cell division and elongation in rice, which is a plant that is also in the Poaceae family [42,88,89,90]. Other studies also found that GA can promote internode elongation by increasing the cell length and cell number of *P. edulis* seedlings during the growth period. Thus, it is reasonable to consider that the promoting effect of exogenous GA on internode length is also achieved by increasing the cell length and number during the rapid growth period.

As an energy source and one of the main storage forms of carbohydrates in plants [25,26], sugar affects plant growth in many ways. Sugar can promote the growth of lateral branches and roots, improve the photosynthetic capacity of leaves, and control the opening and closing of stomata [91,92]. The exogenous Suc treatment significantly promoted internode elongation (Figure 1c–f and Figure 3a). This finding is consistent with those of Wei et al. [37], who found that increasing soluble sugar content inhibited *BmSnf1*, a growth-inhibiting gene, in *Bambusa multiplex* (Lour.) Raeusch. ex Schult, and helped internodes enter into rapid growth because *BmSnf1* is known to be active under low-energy conditions and acts as a growth inhibitor [93]. The sugar content increased exponentially in the division and elongation zones in the internodes during the rapid growth period, indicating that the growth of internodes requires a large amount of sugar [37]. Meanwhile, in a study of *F. yunnanensis*, it was also demonstrated that the starch storage in bamboo shoots could not meet the elongation of the internodes during the rapid growth period, and an additional supply was required from the parental bamboo through rhizomes [36,87,94]. Therefore, we believe that exogenous Suc treatment supplemented the lack of starch and sugar during the rapid growth of *P. edulis* and promoted internode elongation.

We modeled the interaction effects on internode length between exogenous treatments and internode number, analyzing whether the promotion effects of exogenous Suc and GA on internode elongation varied with distance from the injection point. In our results, the effect of the exogenous Suc treatment did not diminish with distance from the injection point, while the effect of the exogenous GA treatment became stronger between the 10th and 20th internodes and became weaker between the 20th and 30th internodes with distance (Figure 4). The same trend can be seen in Figure 3; starting from the injection point, the promoting effect of exogenous GA on the internode length becomes stronger and reaches a peak at around the 25th internode. The GA_3_ used in this study was shown to promote internode elongation in various crops, such as soybeans [95], oats [96], and corn [97]. The promotion of internode elongation by GA_3_ varied with the distance; we believe that because the injected dose was not saturated relative to the demands of the rapid growth period, this led to large consumption of the exogenous GA_3_ near the injection point. These findings support the hypothesis of Chen et al. [65], who barely detected GA_3_ in *P. edulis* shoots during the rapid growth period (18th internode), consistent with our unpublished results. These results may mean that the endogenous GA_3_ in *P. edulis* shoots during the rapid growth period could not meet the demand and was completely depleted. We may be able to use larger doses of GA_3_ to promote internode elongation in the future.

### 3.3. Effects of the Exogenous Treatments on the Internode Numbers

A total of 7728 internodes of 124 individuals of *P. edulis* were measured in the present study. Our results showed that the internode numbers of *P. edulis* were not significantly increased under exogenous GA treatment compared with the CTRL, consistent with Honi et al. [98], who found no significant change in the internode numbers of jute under GA treatment. There are few studies concerning internode numbers, and almost all such studies investigated crops. For example, high temperatures can increase the internode numbers in soybeans [99,100,101], while drought can decrease the internode numbers in sugarcane crops [102]. We only found one report on the internode numbers in bamboo species describing how the internode length was significantly positively correlated with the internode number [67].

Nodes are one of the important characteristics of bamboo culms because nodes will strengthen the culms and prevent the deformation and cracking of internodes [103,104]. The internode numbers under the exogenous Suc treatment were significantly higher than those of the CTRL in the present study, and the average internode number increased by about four (Figure 1b). This result has not previously been reported. The bamboo shoots usually go dormant in the soil and wait for germination [105,106,107]. Bamboo shoots start to differentiate nodes and internodes in the S-4 developmental stages, and internodes are completely differentiated in the S-6 developmental stages [105]. It is generally believed that the internode numbers will not change after the shoots emerge from the soil. Nevertheless, even for the shoots in the ground, the shoot tips are not completely lignified. Is it possible that the supply of Suc promotes the differentiation of the undifferentiated cells in apical meristem? These results must be interpreted with caution because so many factors can affect the development of bamboo in the wild. Our results cannot confirm this conjecture, and we need to sample the bamboo shoots with a finer period division and observe the changes in the apical meristem.

The internode numbers were significantly increased under exogenous Suc treatment, but we also wanted to know whether the added internodes were short or long, as this is important for post-harvest bamboo material processing [16,17,18,19]. Figure 5 depicts the results; compared with the CTRL, the proportion of 30–40 cm internode–s in the exogenous Suc treatment increased significantly and accounted for more than 30% of the total. Although the proportions of 10–20 cm and 20–30 cm internodes were lower under exogenous GA treatment, the proportions of 30–40 cm and 40–50 cm internodes increased significantly. This result suggests that the increase in the internode numbers under Suc treatment could not only help to increase the stability of the bamboo culm but it can also have economic implications in bamboo material processing. In Figure 6, we illustrate the variation in the proportion of internode numbers of different lengths with plant height. The proportion of the shortest internodes (0–10 cm) in the CTRL decreased with the increase in plant height, while there was a slight increase in the exogenous Suc and GA treatments (Figure 6a,f) owing to the internodes being short and thin at the bamboo tip. There were more tall bamboos and more short and thin internodes under the exogenous Suc and GA treatments, and this may have been related to the difficulty in supplying water and nutrients to the higher places we mentioned earlier. The proportion of shorter internodes (10–20 cm and 20–30 cm) decreased with the increase in plant height, and the proportion of longer internodes (30–40 cm and 40–50 cm) increased with the increase in plant height. This trend was stronger under exogenous Suc and GA treatments than in the CTRL (Figure 6b–e,g–j). This coincided with the results shown in Figure 5; taller culms increased the proportion of long internodes and decreased the proportion of short internodes under the exogenous Suc and GA treatments.

An interesting finding is that the intersection of the interaction effects (Figure 6b–e,g–j) was at around 16 m in plant height. This result indicates that once the plant height of *P. edulis* exceeds 16 m, the regulation effects of exogenous Suc and GA on the ratios of long or short internodes begin to weaken, and the rate of return will start to decrease. There are physical limits to the heights of plants; these limits depend on their geographical location, precipitation, physical structure, and water transport capacity [108,109,110,111,112]. For taller bamboo plants (close to the physical limit), even a small increase in height can have a huge potential cost. In contrast, using exogenous Suc and GA on bamboo shoots that should be stunted may obtain a higher rate of return. The exogenous Suc and GA treatments may be more effective in areas where bamboo is not growing well.

### 3.4. Effects of Exogenous Treatments on the Plant Height and Ecological Benefits

The exogenous Suc and GA treatments significantly increased the plant height of *P. edulis* compared with the CTRL (Figure 1a). The plant height of bamboo comprises two parts: the internode length and internode number. The exogenous GA treatment significantly increased the internode length, while the exogenous Suc treatment increased the internode length and internode number (Figure 1b–f). Combined with the visualization results shown in Figure 3, we concluded that the promotion of internode length under exogenous Suc and GA treatments acted on most internodes. In other words, most of the internode elongation by the exogenous treatments resulted in an increase in plant height rather than only on some specific internodes. The internode elongation under the exogenous Suc treatment was weaker than that under the exogenous GA treatment, but the increase in internode number was greater. The internodes have low strength along the grain [103], and the nodes have an important mechanical function on the culm, preventing bending, buckling, and axial cracking [103,104]. This means that the increased nodes under the exogenous Suc treatment may be better from the perspective of culm stability, especially in some mountainous areas with harsh terrain. However, considering the utilization of bamboo material, the increased internode length under exogenous GA treatment has greater economic value; these two methods should be used differently depending on the occasion.

Although the exogenous Suc and GA treatments provided positive results in this study, considering the physiological integration ability of bamboos [113,114,115], we still believe that the promotion effect was underestimated. This is because when the exogenous treatment supplies the starch or hormones that the bamboo needs during the rapid growth period, the parent plant may redistribute its extra resources to another bamboo through the rhizomes (which may have included the control in the experiment). The results shown in Figure 1a also demonstrate the following: there were fewer bamboos with lower plant height under the exogenous Suc and GA treatments, and the data points were more concentrated compared with the control, suggesting that the exogenous treatments helped some *P. edulis* shoots that had limited growth potential to grow taller. If these samples were taken out of the data, the effects of the exogenous Suc and GA on the plant height and internode length might have been greater than what is shown in the results. In future studies, it may be necessary to set up an enclosed area or perform rhizome truncation treatments [113,114,116], and reevaluate the effect of exogenous Suc and GA treatments.

China plants a large amount of bamboo yearly, and bamboo can accumulate 76% of the biomass over the next five years during the rapid growth period [70,117]. The average carbon stock of *P. edulis* forests is 2.39 times higher than that of China fir (*Cunninghamia lanceolata*) forests, and thus, has a huge carbon sequestration potential [70,117,118,119,120]. Our results demonstrated that the biomass and carbon storage per plant of *P. edulis* increased significantly under the exogenous Suc and GA treatments, with average increases of 5.9 kg and 13.0 kg in biomass and average increases of 2.8 kg and 6.3 kg as a carbon sink (Figure 7). Although the plant height and diameter of bamboo cannot increase after the rapid growth period [117], a higher culm can support more branches and leaves. Considering the ecological benefits, the effects of exogenous Suc and GA treatments on plant height are meaningful until bamboo felling. The felled bamboo is processed into building materials [7,121,122,123], which is a process that ensures that this part of the sequestered carbon will not return to nature within a short time. The global forest area has continued to decline over the past 30 years, but bamboo forests have grown at an average annual rate of 3% [124]. *P. edulis* is China’s most widely planted bamboo species [12]. Using low-cost Suc or GA to enhance *P. edulis* can increase the output value and help bamboo forests play an important role in mitigating climate change.

In the context of bamboo processing, the increase in internode length evidently contributes to the enhancement of economic benefits, as longer internodes allow for the production of a greater number of bamboo products. For bamboo handicrafts such as fans, walking sticks, and furniture, an increase in the length of individual internodes can significantly elevate the added value of the products. However, to quantify the impact of increased internode length on the economic benefits of these products, it will be necessary to gather more relevant data in future work, such as the price variations of walking sticks of different lengths.

The present study is the first to verify the effects of exogenous Suc and GA treatments on the plant height of *P. edulis* in the field. Because many bamboo species have similar physiological and ecological characteristics [125,126], we have reason to believe that the promoting effects of Suc and GA on the internode length and internode numbers may not be limited to *P. edulis*, but may also be effective for other bamboo species. This low-cost method of increasing the plant height and internode length may be highly profitable in some areas where bamboo is not growing well, both economically and ecologically. In future research, we will try to remove the effect of physiological integration and increase the number of bamboo species and hormone types.

## 4. Materials and Methods

### 4.1. Study Site and Experiment Design

In the present study, all experimental subjects were selected from pure stands of *P. edulis* located in Guangde City of Anhui Province (119.384 E, 30.799 N), Yanling County of Hunan Province (114.033 E, 26.548 N), and Xianning City of Hubei Province (114.313 E, 29.815 N) in the south of China (Figure 8). The soils were predominantly yellow loam; the temperatures and precipitation (Appendix A) were suitable for *P. edulis* growth as a major distribution area in China (annual average temperature: 15.6–19.5 °C, annual total precipitation: 1436.6–1876.2 mm) [127]. We randomly labeled 160 bamboo shoots at a height of close to 50 cm with a similar ground diameter and recorded the ground diameter of all shoots in each study site at the end of March 2021. We injected 50 mL of 20 μM Suc (Shanghai Yuanye Bio-Technology Co., Ltd., Shanghai, China; S11055) and 20 μM GA_3_ (Shanghai Yuanye Bio-Technology Co., Ltd., Shanghai, China; S28506) solutions into 60 shoots, and as a control, we injected 50 mL of pure water into another 60 shoots in each study site. We did not perform the exogenous treatment via spraying because the solution is usually sprayed on the leaves; however, bamboo grows very quickly during the rapid growth period and can reach heights that cannot be sprayed in a short time. The injection location was the fifth internode from the bottom. The injection was carried out every three days, five times in total in each study site from 28 March 2021 to 16 April 2021. Parenthetically, we found the suitable dose of Suc and GA to increase the plant height of bamboo shoots via pre-experiments from 2019 to 2020 in Guangde City, Anhui Province (119.384 E, 30.799 N). Virtually all bamboo shoots stopped their rapid growth by September. Hence, we measured the plant heights of the bamboo (*n* = 241) using the criterion RD 1000 (Laser Technology, USA) at the end of August 2021, except for other degraded bamboo shoots. Meanwhile, we felled 124 bamboo shoots (CTRL: 36; Suc: 62; GA: 26) and measured the internode length after labeling every internode. Owing to local laws and manpower constraints, we were not able to fell more bamboo plants.

### 4.2. Statistical Analysis

We analyzed the relationships between variables using a generalized linear mixed model (GLMM) with the Bayesian Markov chain Monte Carlo (MCMC) method. There are many advantages of Bayesian modeling approaches, including having the flexibility to set the distribution family of the models according to the characteristics of the data and fitting the data well even with small sample sizes. Therefore, the Bayesian approach is used in a wide range of applications [128,129,130]. In the present study, the internodes were divided into 10th–20th, 20th–30th, 30th–40th, and 40th–50th ranges for analysis because in the culm morphology of *P. edulis*, the closer to the tip, the thinner the culm. The 1st–10th internodes near the base and after the 50th internodes are shorter in length and have fewer applications. We set the plant height, internode number, internode length, and percentage of internodes with different lengths, estimated biomass, and carbon storage per culm as the dependent variables, with exogenous treatment, distance from the injection point, and plant height as the independent variables for modeling. The details of all model structures are shown in Table 1. For each model, we set 4 independent chains with 1 interval and obtained 5000 posterior samples from each chain after a burn-in of 5000 iterations. Prior distributions of all models used default settings in the brms package 2.17.0 [131]. We considered that zero not being within the 95% credible interval of variables indicated a statistically significant result. We calculated the R-hat values of all parameters and checked that weather was less than 1.01 [132]. 

The above-ground biomass and carbon storage per culm were calculated using the formulas below [133]:(1)$$B_{culm}=\frac{2t+\frac{R}{t}+8}{100t+20\frac{R}{t}-920}\times H_{culm}D^{\left[150825-564.8t+98.6\frac{R}{t}\right]10^{-5}}$$
(2)$$C_{culm}=B_{culm}\times cc_{culm}\times 100\%$$
where *B_culm_* is the above-ground biomass per culm, *t* is the annual average temperature, *R* is the annual average precipitation, *H_culm_* is the culm height, *D* is the ground diameter, *C_culm_* is the carbon storage per culm, and *cc_culm_* is the carbon content of the *P. edulis* culm [134].

Analyses and visualizations were done using R v4.2.0 and ArcMap v10.2.1.

## 5. Conclusions

To the best of our knowledge, this study demonstrated for the first time that the plant height of *P. edulis* was promoted by regulating the co-elongation of most internodes and internode numbers during the rapid growth period. To address the questions in the introduction, we drew four main conclusions: (1) exogenous GA affected the internode length, while exogenous Suc affected the internode length and internode numbers; (2) most of the internode elongation resulted in the increase of plant height rather than only on some specific internodes; (3) the promotion of internode elongation using exogenous GA decreased with distance; and (4) the promotion of plant height using exogenous Suc and GA was better for bamboos below 16 m. Because many bamboo species have similar physiological and ecological characteristics, we have reason to believe that these insights may not be limited to *P. edulis*, but may also be effective for other bamboo species. These conclusions are helpful to understand the physiological and morphological characteristics of bamboo culms during the rapid growth period and provide a low-cost way to improve the economic and ecological benefits of Moso bamboo forests.

## Figures and Tables

**Figure 1 plants-12-01713-f001:**
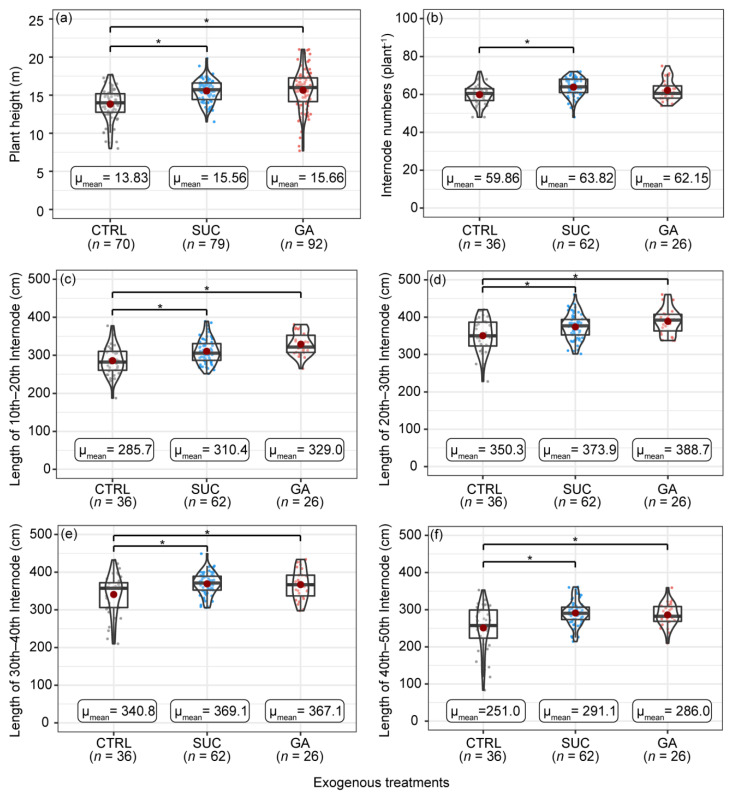
Comparisons of the plant height (**a**), internode number (**b**), lengths of the 10th–20th internodes (**c**), lengths of the 20th–30th internodes (**d**), lengths of the 30th–40th internodes (**e**), and lengths of the 40th–50th internodes (**f**) under different exogenous treatments. Box plots and violin plots show the data distribution of each variable. Points represent the measured values of each variable. Red points indicate mean values. The numbers in brackets represent sampling numbers. An asterisk means a significant difference according to the Bayesian GLMM.

**Figure 2 plants-12-01713-f002:**
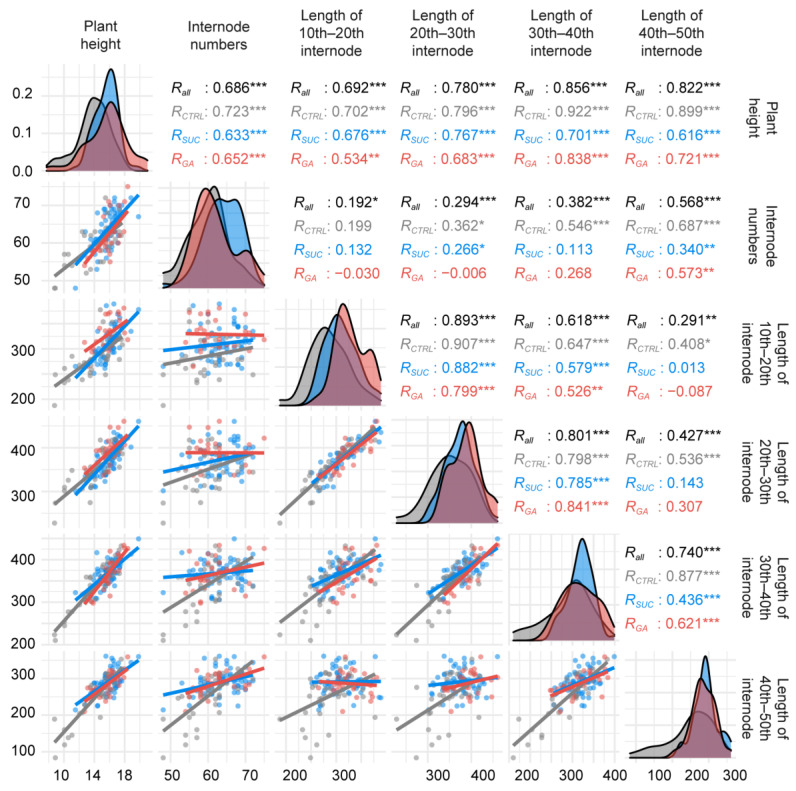
Correlations between the plant height; internode numbers; and lengths of the 10th–20th internodes, 20th–30th internodes, 30th–40th internodes, and 40th–50th internodes under different exogenous treatments. The different colors show each exogenous treatment. Numbers indicate Pearson correlation coefficients (* *p* < 0.05, ** *p* < 0.01, *** *p* < 0.001). Multidensity plots and scatterplots show the distributions of the variables.

**Figure 3 plants-12-01713-f003:**
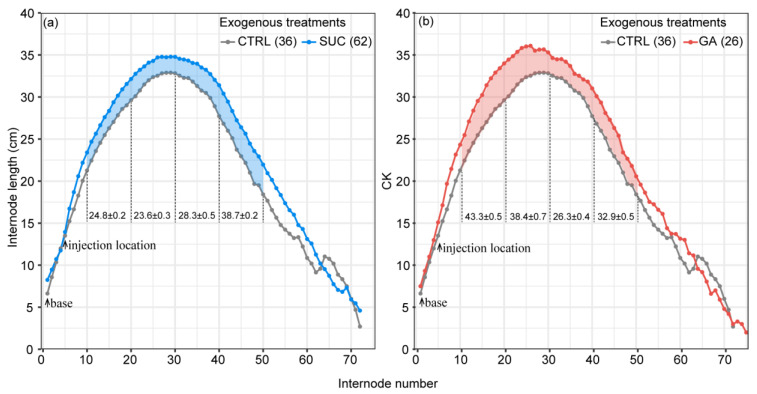
The internode lengths at different internodes under Suc (**a**) and GA (**b**) treatments compared with the CTRL. The points indicate the mean value of the internode length. The colored shapes indicate a significant difference in the length of every ten internodes between the Suc or GA treatment and the CTRL. The mean values and standard error of 10 internodes with a significant difference are shown between the lines. The numbers in brackets represent the sample sizes.

**Figure 4 plants-12-01713-f004:**
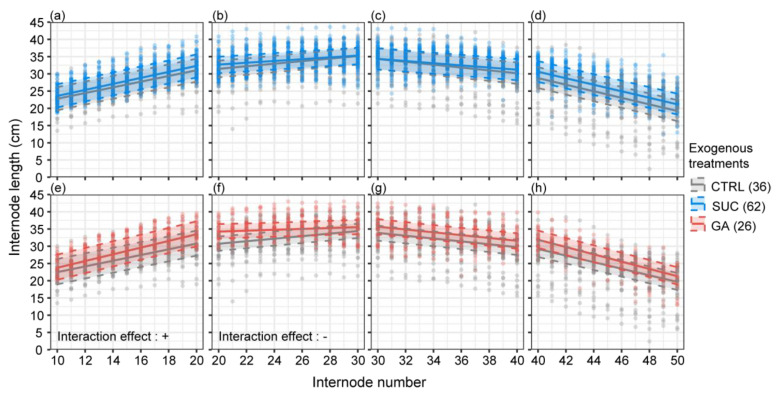
The effect of the interaction between exogenous treatments and internode number on the internode length. The points represent the measured values of the internode length. The shaded intervals within the dashed lines show the 95% Bayesian credible intervals. The numbers in brackets represent the sample sizes (**a**–**h**). The symbols + or − mean a significant positive interaction or negative interaction, respectively.

**Figure 5 plants-12-01713-f005:**
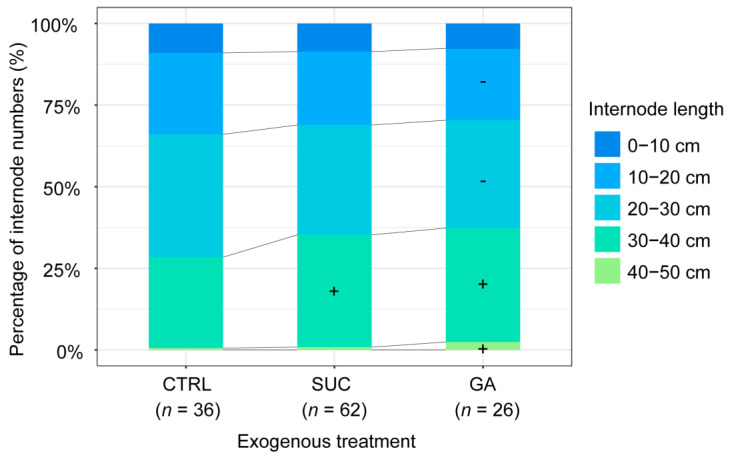
The percentages of internodes with different lengths under different exogenous treatments. The numbers in brackets represent the sample sizes. The symbols + or − mean a significant increase or significant decrease in the proportion of internodes compared with the CTRL, respectively.

**Figure 6 plants-12-01713-f006:**
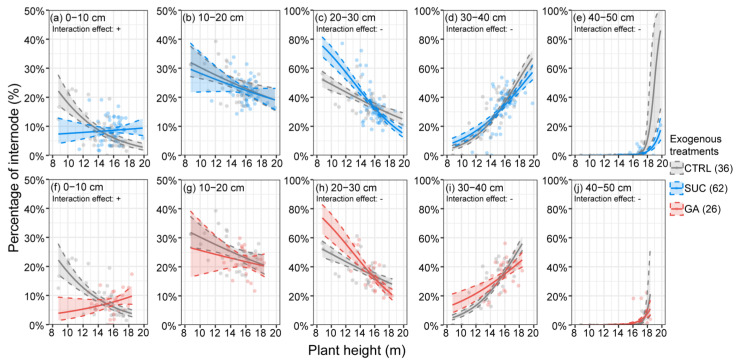
The effects of the interactions between the exogenous treatments and plant height on the percentage of internodes with different lengths ((**a**,**f**) 0−10 cm, (**b**,**g**) 10−20 cm, (**c**,**h**) 20−30 cm, (**d**,**i**) 30−40 cm, and (**e**,**j**) 40−50 cm). The points represent the percentage of internodes of each length at different plant heights. The shaded intervals within the dashed line show the 95% Bayesian credible intervals. The numbers in brackets represent the sample sizes. The symbols + or − mean a significant positive or negative interaction, respectively.

**Figure 7 plants-12-01713-f007:**
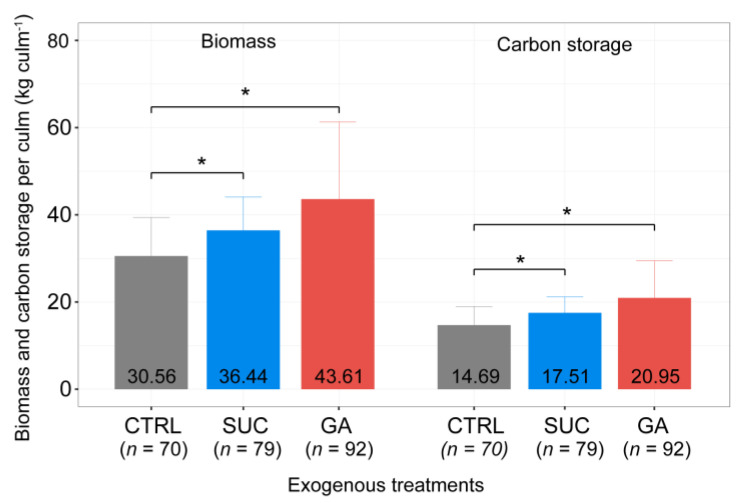
Comparison of the estimated biomass and carbon storage under different exogenous treatments. The numbers at the bottom of the bar graph represent the average values. The error bars show the 95% Bayesian credible intervals. The asterisk means a significant difference according to the Bayesian GLMM. The numbers in brackets represent the sample sizes.

**Figure 8 plants-12-01713-f008:**
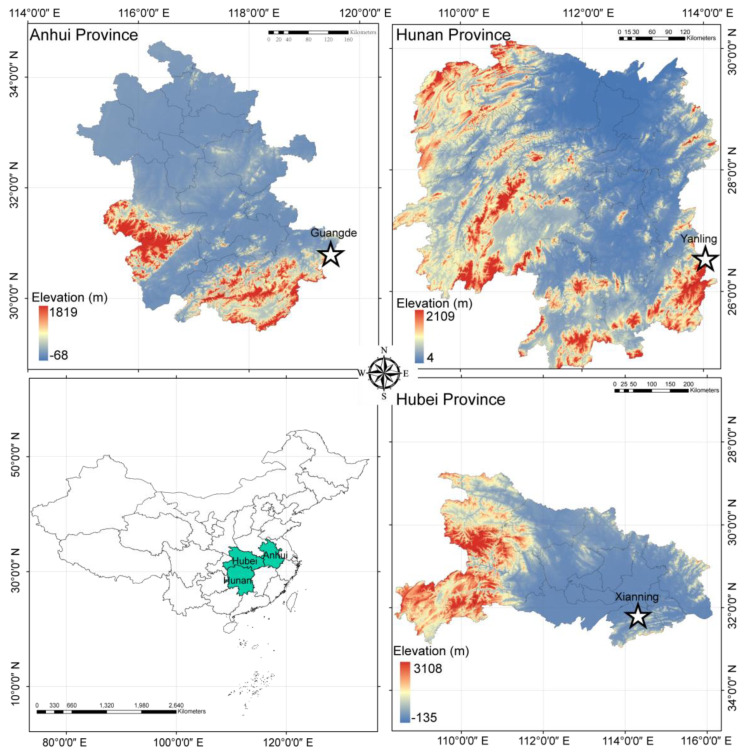
Study area.

**Table 1 plants-12-01713-t001:** The structure of the Bayesian GLMM.

Model Structure	*i*	*j*	*k*	*l*
*Ph_i_* or *Eb_i_* or *Ec_i_* ~ *Normal* (*μ_i_*, *σ*^2^*_i_*)	1, 2, …, 241	0 or 1		1, 2, 3
*μ_i_* = *β*_0_ + *β*_1_*S_j_* + *β*_2_*G_j_* + *r*_1*i*_ *+ r*_2*l*_				
*In_i_* ~ *Poisson* (*λ_i_*)	1, 2, …, 124	0 or 1		1, 2, 3
*log*(*λ_i_*) = *β*_0_*+ β*_1_*S_j_ + β*_2_*G_j_ + r*_1*i*_*+ r*_2*l*_				
*Il*^10–20th or 20–30th or 30–40th or 40–50th^*_i_* ~ *Normal* (*μ_i_*, *σ*^2^*_i_*)	1, 2, …, 124	0 or 1		1, 2, 3
*μ_i_* = *β*_0_ + *β*_1_*S_j_* + *β*_2_*G_j_* + *r*_1*i*_ *+ r*_2*l*_				
*Il*^10–20th or 20–30th or 30–40th or 40–50th^*_i_* ~ *Normal* (*μ_i_*, *σ*^2^*_i_*)	1, 2, …, 124	0 or 1	1, 2, …, 75	1, 2, 3
*μ_i_* = *β*_0_ + *β*_1_*S_j_* + *β*_2_*G_j_* + *β*_3_*D_k_* + *β*_4_*S_j_D_k_* + *β*_5_*G_j_D_k_* + *r*_1*i*_ *+ r*_2*l*_				
*In*^0–10 cm or 10–20 cm or 20–30 cm or 30–40 cm or 40–50 cm^*_i_* ~ *Binomial* (*In_i_*, *logistic* (*p_i_*))	1, 2, …, 124	0 or 1		1, 2, 3
*p_i_* = *β*_0_ *+ β*_1_*S_j_ + β*_2_*G_j_ + r*_1*i*_ *+ r*_2*l*_				
*In*^0–10 cm or 10–20 cm or 20–30 cm or 30–40 cm or 40–50 cm^*_i_* ~ *Binomial* (*In_i_*, *logistic* (*p_i_*))	1, 2, …, 124	0 or 1		1, 2, 3
*μ_i_* = *β*_0_ + *β*_1_*S_j_* + *β*_2_*G_j_* + *β*_3_*Ph_i_* + *β*_4_*S_j_Ph_i_* + *β*_5_*G_j_Ph_i_* + *r*_1*i*_ *+ r*_2*l*_				

*Ph_i_*: plant height of culm *i*; *Eb_i_*: estimated biomass of each culm *i*; *Ec_i_*: estimated carbon storage of each culm *i*; *In_i_*: internode numbers of each culm *i*; *Il*^10th–20th^*_i_*: 10th–20th internoder length of each culm *i*; *Il*^20th–30th^*_i_*: 20th–30th internode length of each culm *i*; *Il*^30th–40th^*_i_*: 30th–40th internode length of each culm *i*; *Il*^40th–50th^*_i_*: 40th–50th internode length of each culm *i*; *In*^0–10 cm^*_i_*: 0–10 cm internode numbers of each culm *i*; *In*^10–20 cm^*_i_*: 10–20 cm internode numbers of each culm *i*; *In*^20–30 cm^*_i_*: 20–30 cm internode numbers of each culm *i*; *In*^30–40 cm^*_i_*: 30–40 cm internode numbers of each culm *i*; *In*^40–50cm^*_i_*: 40–50 cm internode numbers of each culm *i*; *β*: intercept or regression coefficient for each variable; *r*_1*i*_: random effect of ground diameter in each culm *i*; *r*_2*l*_: random effect of each study site *l*; *S*: exogenous SUC treatment for each culm *i*; G: exogenous GA treatment for each culm *i*; D: distance from the jection point of each culm *i*.

## Data Availability

Not applicable.

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
