# Peer review of "Plasticity in the Morphology of Growing Bamboo: A Bayesian Analysis of Exogenous Treatment Effects on Plant Height, Internode Length, and Internode Numbers"

_plants, 2023, doi:10.3390/plants12081713_

Round 1
Reviewer 1 Report
In the submitted manuscript the authors evaluated the effects of exogenous Sucrose and Gibberellin treatments on the plant height, internode length, and internode number in a bamboo forest. The issue is interesting, especially from an economic and ecological point of view.
Overall, the topic is interesting and the results are convincing and well argued in the Discussion section. However, there are a few minor issues that needed to be addressed before publication.
1. Please, inspect all the abbreviations. They should be defined the first time they appear in the text.
2. I suggest rewriting the abstract to highlight the importance of the study. In some points, it is repetitive and lacks clarity.
3. In the first paragraph of the Results section, numeric data should be organized in a table in order to improve the smoothness of the text.
4. Experimental planning is not clear to me. The number of analyzed samples is lower than those effectively analyzed in the results. Please, explain it.
5. Data were collected in three different areas, I think would be interesting to compare the results obtained in the different sites to check possible differences.
Reviewer 2 Report
The authors present an interesting report on the promoting effect of exogenous Suc and GA on culm height, internode length, and internode number of Moso bamboo in the field. A very detailed analysis of the relationship between culm height, internode number and internode length under different treatments was conducted. Even more impressive was that the internode length of the 10thto 50th internodes was significantly longer and the proportion of 30-40 cm internodes was significantly higher under the treatments with exogenous Suc and GA. This could be very useful to increase the economic and ecological benefits of bamboo forest through a cost-effective method. The extensive field work, detailed statistics of individual internode lengths and numbers, and scientific rigour of the analysis are credible and worth reading. The results provide new information for further understanding the plasticity of clump morphology in exogenous treatments in woody bamboo. However, there are some problems with the publication. The following issues could be considered for revision.
1. Introduction
-Line 98 Change ‘Gibberellic’ to ‘gibberellic’, and ‘General’ to ‘general’
2. Materials and methods
- Line 454-457: The experiment was conducted at three study sites. Although the map (Fig. 8) shows that the three study sites are adjacent to each other and there is not much difference in altitude, the authors did not provide information on meteorological conditions and soil nutrients at the three different sites. Will these factors affect the current results?
3. Results
- Lines 158-163, 167-171, 193-197, 199-202, 215-217, 236-241: In the manuscript, the total bamboo was divided into 5 sections of 10-20, 20-30, 30-40, and 40-50 for analysis (Figure 1-6). What is the reason why other internodes were not included in the analysis?
- Line 465-466: Remove the extra "-"
Reviewer 3 Report
This article describes the use of GA3 and sucrose to improve bamboo production. The physiological part is well done and carefully analyzed. However, this reviewer believes that biochemical parameters should be incorporated to support the conclusions reached by the authors. For example, it is said that the effect is more effective if the plants do not reach heights greater than 16 m. A simple GA3 determination could be made throughout the plant to see how the injected GA3 has been distributed. In principle, there should be a relationship between the GA3 concentration and the observed effect. Same for injected sucrose. The water potential could also be measured to support the data that the hydraulic pressure from the root is insufficient to move the water. One unanswered question is whether the plant's photosynthesis changes once sucrose is injected.
One last comment about the manuscript is that it is excessively referenced. Some paragraphs have up to 4 or 5 references to document or support what was said. This reviewer thinks that in most cases, two citations are acceptable, maximum of three references.
Some specific comments are listed below, and the line number in which they are found.
Line 71. Wang et al. (2020) is not in the format for references of the journal.
Line 74. Wei et al. (2019) is not in the format for references of the journal.
Line 100. Wang et al. (2018) is not in the format for references of the journal.
Lines 154, 170. The letter P for statistic must be P. Please correct trough the manuscript.
Line 330. Chen et al. (2022) is not in the format for references of the journal.
Line 461. Please provide the catalog number, company, and country for the reagents.
Line 462. CK is, in general, used as an abbreviation for cytokinin. Please avoid the use of CK as an abbreviation for control.
The language of the manuscript is good. This reviewer recommends small changes in the redaction of the paper as those illustrated in the abstract.
Abstract: Suc and GA can promote the elongation of certain internodes in bamboo. However, there is no evidence concerning how Suc and GA promote the plant height of bamboo by regulating internode elongation and number. We investigated the plant height, the length of each internode, and the total number of internodes of P. edulis under exogenous Suc, GA, and CK treatments in the field. and We analyzed how Suc and GA affected the height of P. edulis by promoting internode length and number. The lengths of the 10th–50th internodes were significantly increased under exogenous Suc and GA treatments, and the number of internodes was significantly increased by exogenous Suc treatment. The proportion of 30–40 cm internodes was significantly increased under exogenous Suc treatment; the proportion of 30–50 cm internodes was significantly increased, and the proportion of 10–30 cm internodes was significantly decreased under exogenous GA treatment. This study demonstrated that both exogenous Suc and GA treatments could promote internode elongation of P. edulis in the field. Exogenous GA treatment had a stronger effect on internode elongation, and exogenous Suc treatment had a stronger effect on increasing internode numbers. The co-elongation of most internodes or the increase in the proportion of longer internodes promoted the increase in plant height by exogenous Suc and GA treatmentsThe increase in plant height by exogenous Suc and GA treatments was promoted by the co-elongation of most internodes or the increase in the proportion of longer internodes.
Reviewer 4 Report
The manuscript by Wu et al. is devoted to the study of the effect of the exogenous sucrose and gibberellin on the Phyllostachys edulis height and internode parameters. This work contains very interesting results, but I have comments:
1. Why is there an emphasis on Bayesian analysis in the title? Is it important? If so, why is nothing written about it in the Abstract?
2. The Abstract section contains many details, but does not contain a general conclusion.
3. P. 2, lines 54-55: It is appropriate to add here what is currently used to increase the height of woody bamboo or what could potentially be used besides sucrose and gibberellin.
4. P. 10, line 306: “BmSnf1” needs decryption.
5. Lost words at the end of the Conclusion section?
Reviewer 5 Report
In the manuscript "Plasticity in Morphology of Growing Bamboo: A Bayesian Analysis of Exogenous Treatment Effects on Plant Height, Internode Length and Internode Numbers", the authors reported that shoot injection treatments with sucrose or gibberellin are highly recommended for increasing P. edulis plant height, and internode length and number. It is an interesting and clear study with a valid and better selection of samples. The study included well-presented data and analysis, and tables and figures are clarified. However, minor revisions are needed as follows:
- Supplementary Tables: The tables are deficient in the descriptions of treatments. Please add treatment descriptions in the footnotes.
- Line 11: Please write the full name when it is mentioned for the first time, and then write the abbreviation.
- Line 26: Instead of "Exogenous Treatment", write what is used for the exogenous treatment.
- Lines 41-42: The authors mentioned that "In 2020, the total import and export trade of bamboo 41 products in China was 2.21 billion US dollars", what about 2021-2022?
- Line 256: The authors should not use their own pronoun; "we". Please check also for the entire manuscript.
- At the end of the discussion section, can the authors write some lines elucidating the economic financial profitability resulting from the applied treatments, and are these applied treatments economically feasible or not?
- Line 458: Please write the temperatures and precipitation that are suitable for P. edulis growth in the study area.
- Line 461: What are the volumes of 20 μM Suc and 20 μM GA3 injected into P. edulis shoots? Are they 50 mL like the volume of water? And the water injected into plant shoots is it distilled or what?
Reviewer 6 Report
This is an original and interesting study that investigates the plant height, internode length, and internode numbers of Phyllostachys edulis under exogenous sucrose and gibberellin treatments. The authors found that this low cost method of increasing plant height and internode length may be highly profitable in some areas where bamboo is not growing well, both economically and ecologically.
In my opinion, the manuscript presented for review is very well prepared. I have no substantive comments, I believe that the manuscript is refined and contains interesting results and appropriate conclusions. The study's methodology can be useful for future research and development in the field. The discussion section presents a good comparison of the obtained results with other results available in the data basis. There is very accurate and well-chosen literature for discussion and comparison. Conclusion section is well formulated and corresponding with the obtained results.
After carefully manuscript reading, I think, that presented experiment is valuable. However, I have listed some suggestions for correction that the authors must consider.
L. 11. I suggest you write the full form of the words Suc and GA when they appear for the first time in text and put the abbreviation in brackets. After that, you can use the abbreviated form in the continuation of the text. Because, if someone wanted to read only the abstract, they wouldn't know what those abbreviations mean. The same suggestion applies to CK.
L. 27. You must not use the same keyword twice. So you can remove one of the words Internode.
L. 30. Write the scientific name in italics. Please check the entire manuscript.
L. 457-458. I think you should provide more details about both the soil and the climatic conditions in the experimental areas.
